# COVID-19 Biomarkers Comparison: Children, Adults and Elders

**DOI:** 10.3390/medicina59050877

**Published:** 2023-05-03

**Authors:** Ionut Dragos Capraru, Dan Dumitru Vulcanescu, Iulia Cristina Bagiu, Florin George Horhat, Irina Maria Popescu, Luminita Mirela Baditoiu, Delia Muntean, Monica Licker, Matilda Radulescu, Ion Cristian Mot, Mircea Mihai Diaconu, Catalin Marian

**Affiliations:** 1Department of Epidemiology, “Victor Babes” University of Medicine and Pharmacy, Eftimie Murgu Sq. Nr.2, 300041 Timisoara, Romania; ionut.capraru@umft.ro (I.D.C.); irina.stefan@umft.ro (I.M.P.); baditoiu.luminita@umft.ro (L.M.B.); 2Department of Microbiology, Multidisciplinary Research Center on Antimicrobial Resistance, “Victor Babes” University of Medicine and Pharmacy, Eftimie Murgu Sq. Nr.2, 300041 Timisoara, Romania; dan.vulcanescu@umft.ro (D.D.V.); horhat.florin@umft.ro (F.G.H.); muntean.delia@umft.ro (D.M.); licker.monica@umft.ro (M.L.); radulescu.matilda@umft.ro (M.R.); 3Clinical Laboratory, Emergency Hospital for Children “Louis Turcanu”, 300011 Timișoara, Romania; 4Microbiology Laboratory, “Pius Brinzeu” County Clinical Emergency Hospital, No. 156 L. Rebreanu, 300723 Timisoara, Romania; 5ENT Department, “Victor Babes” University of Medicine and Pharmacy, Eftimie Murgu Sq. No. 2, 300041 Timisoara, Romania; cristianmotz@yahoo.com; 6Department of Obstetrics and Gynecology, “Victor Babes” University of Medicine and Pharmacy, 300041 Timisoara, Romania; diaconu.mircea@umft.ro; 7Department of Biochemistry, “Victor Babes” University of Medicine and Pharmacy, Eftimie Murgu Sq. Nr.2, 300041 Timisoara, Romania; cmarian@umft.ro

**Keywords:** COVID-19, SARS-CoV-2, biomarkers, children, adults, elders

## Abstract

*Background and Objectives:* this study aimed to research links between C-reactive protein (CRP), lactate dehydrogenase (LDH), creatinekinase (CK), 25-OH vitamin D (25-OHD), ferritin (FER), high-density lipoprotein cholesterol (HDL)cholesterol and clinical severity in patients from the western part of Romania, and compare their potential use as biomarkers for intensive care units (ICU) admission and death in children, adults and elders. *Materials and Methods:* this study is a retrospective cohort study, performed on patients positively diagnosed with COVID-19. Available CRP, LDH, CK 25-OH vitamin D, ferritin, HDL cholesterol and clinical severity were recorded. The following were assessed: median group differences, association, correlation and receiver operating characteristic. *Results:* 381 children, 614 adults and 381 elders were studied between 1 March 2021 and 1 March 2022. Most children and adults presented mild symptomatology (53.28%, 35.02%, respectively), while most elders presented severe symptomatology (30.04%). ICU admission was 3.67% for children, 13.19% for adults and 46.09% for elders, while mortality was 0.79% for children, 8.63% for adults and 25.1% for elders. With the exception of CK, all other biomarkers showed some significant associations with clinical severity, ICU admission and death. *Conclusions:* CRP, LDH, 25-OH vitamin D, ferritin and HDL are important biomarkers for COVID-19 positive patients, especially in the pediatric population, while CK was mostly within normal ranges.

## 1. Introduction

From 2003 to the present, a variety of coronaviruses have been identified in humans and animals, according to the literature. This kind of virus can easily spread from one person to another, resulting in infections of varying degrees, most commonly the common cold but also more severe infections [1,2]. 

At the end of 2019, a new coronavirus, initially called 2019-nCoV, and later renamed SARS-CoV-2, was recognized in China, causing the COVID-19 disease. It was proclaimed a worldwide pandemic by the WHO (World Health Organization) as of March 2020 [2]. The severe acute respiratory syndrome caused by COVID-19 infection resulted in the admission of numerous patients, particularly elderly people and those with comorbid conditions like diabetes or heart or lung conditions, to intensive care units (ICUs) [3,4].

The inflammatory response in COVID-19 was linked to the following biomarkers: lactate dehydrogenase (LDH), a potential marker of vascular permeability in immune-mediated lung lesions, associated with cardiopulmonary diseases, myocardial infarction, cancer, and other diseases [5]; C-reactive protein (CRP), used for defining inflammatory conditions and injuries [6,7]; creatinekinase (CK), which is associated with mortality [8]; 25-OH vitamin D (25-OHD), which induces cathelicidins and defensins with the ability to slow down viral replication and the production of cytokines that fight inflammation [9]; ferritin (FER), which can show secondary hemophagocytic lymphohistiocytosis, which is associated with poor prognosis and can predict the worsening of COVID-19 patients [10]; and high-density lipoprotein cholesterol (HDL), which has an inverse relation with disease severity, and can decrease disease susceptibility [11].

In a study of adult patients, Dogan Akdogan and his colleagues also demonstrated that both LDH and CRP were associated with lung lesions in the early stages of COVID-19 disease. This association may also indicate the need for chest CT (Computed Tomography) and the severity of the disease [2]. A two-month retrospective study at Wuhan Children’s Hospital in early 2021 on the CK marker came to the conclusion that elevated serum CK values could indicate organ lesions and an increased immune response [12,13].

The deficit of 25-OH vitamin D is a major health issue; it is estimated that over a billion people worldwide lack 25-OH vitamin D [14]. Although there were few studies in the beginning, a link between symptomatology, severity and outcome in COVID-19 with regard to 25-OH vitamin D has been observed [15,16,17].

Hyperferritinemia, caused by excessive inflammation, is associated with admission to the intensive care unit and high mortality, and represents an indication to recognize high-risk patients. It is also present in patients with comorbidities [18,19].

Cholesterol can indirectly increase the susceptibility of patients to SARS-CoV-2 and increase the risk of death. Significantly decreased HDL levels in patients with COVID-19 were correlated with both disease severity and mortality [11,20,21]. A study from Turkey even considers it a good predictor of one-year mortality in patients with COVID-19 [22].

Epidemiological data showed that the COVID-19 disease affected fewer children than adults, although these numbers could be influenced by other factors [23]. However, elders have been observed to be at even a higher risk of severe symptoms and even death [24].

As such, a comparative study between children, adults and elders would deepen our understanding in how these biomarkers are affected by the SARS-CoV-2 in those age groups.

## 2. Materials and Methods

### 2.1. Study Design and Participants

This retrospective cohort study was performed on children, adults and elders positively diagnosed with COVID-19 at a RT-PCR molecular test (Reverse Transcription Polymerase Chain Reaction). This is a multicentric study, as children were admitted to the Emergency Reception Unit (ER) at the Pediatric Hospital “Louis Turcanu”, while the adults and elders were admitted to the ER of the County Emergency Hospital “Pius Brinzeu”. Both hospitals are in Timisoara, Romania, and the study period was between 1 March 2021 and 1 March 2022 (1 year). Data was obtained from the hospitals’ registries and no personal data was recorded. The following ethics approval was obtained from the Pediatric Hospital “Louis Turcanu”: 98/20.01.2022.

The criteria for admission of patients to the study were: a positive COVID-19 RT-PCR test, existing test results for CRP, LDH, CK, 25-OH vitamin D, ferritin and/or HDL. Patients excluded from the study were COVID-19 positive patients without CRP, LDH, CK, 25-OH vitamin D, ferritin and/or HDL data. Pediatric patients were defined as younger than 18 years, while elders were defined as older than 65, with adults being in-between.

The severity of the disease was classified as asymptomatic, mild, moderate, severe and critical according to the clinical characteristic and laboratory results: asymptomatic—cases with a positive RT-PCR test and without any clinical findings; mild—cases with upper respiratory tract infection symptoms, such as fever, fatigue, myalgia, cough, sore throat; moderate—pneumonia with complaints of fever and cough but without the symptoms of dyspnea and hypoxemia; severe—cases with fever and cough in the early period which develop dyspnea and central cyanosis within a week (arterial oxygen saturation of <92%); critical—cases which develop acute respiratory distress or respiratory failure rapidly, and who tend to develop shock, encephalopathy, myocardial affection, coagulation dysfunction and acute kidney injury.

### 2.2. Data Collection

The study has a retrospective cohort character, so the patient’s written consent was not required, as the data were obtained from the computer register of the laboratories. Epidemiological information such as age, sex and diagnosis were also assessed. Identification data of the patients such as names, ID numbers or contact information were not collected for the present study.

The determination of the biomarkers, CRP, LDH, CK and 25-OH vitamin D was performed upon admission to the ER and compared with the corresponding normal values for each biomarker, as follows: CRP (<5 mg/L); LDH (0–1 year: 225–600 U/L, 1–17 years: 120–300 U/L, >17 years: 135–225 U/L); CK (0–4 years: 24–228 U/L, 4–7 years: 24–149 U/L, F: 7–17 years: 24–154 U/L, >17 years: 24–192 U/L, M: 7–17 years: 24–270 U/L, >17 years: 24–308); 25-OHD (>30 ng/mL); FER (0–1 year: 12–327 ng/mL, 1–7 years: 4–67 ng/mL, F: 7–17 years: 7–84 ng/mL, >17 years: 13–150 ng/mL, M: 7–17 years: 14–152 ng/mL, >17 years: 30–400 ng/mL); HDL (>40 mg/dL). Data on the COVID-19 results obtained from laboratory tests were also reviewed.

### 2.3. Laboratory Testing

A fully automated clinical chemistry instrument Cobas Integra 400 plus (Roche Diagnostics, Bucharest, Romania) was used for analyzing the serum sample. Nasopharyngeal swabs of the patients were examined using the BIO-RAD CFX96 Real-Time System C1000 Touch Thermal Cycler Device.

### 2.4. Statistical Analysis

Sample size was calculated using the G*Power software, v 3.1.9.6 (Heinrich-Heine-Universität Düsseldorf, Düsseldorf, Germany), using an a priori test to calculate the minimum sample size for a medium effect size (d = 0.25), a power of 80% and an alpha of 0.05. The result indicated that a minimum of 270 patients were needed.

Statistical analysis was performed using the MedCalc Statistical Software, version 20.218 (MedCalc Software bv, Ostend, Belgium) All continuous variables were assessed with the Kolmogorov–Smirnov test, to check for normal distribution. Continuous variables with a normal distribution were expressed as mean and standard deviation (SD), and were compared using the ANOVA with Tukey’ post hoc test. Non-parametric variables were expressed as median and interquartile range (IQR), and were compared using the Kruskal–Wallis with Dunn’s post hoc tests.

In order to find associations between the existence of clinical respiratory signs and the elevated status of the studied biomarkers, contingency tables were created and analyzed using the Chi^2^ test. Afterwards, links between biomarkers levels were assessed using Spearman’s rank correlation, where r ≤ 0.10 was considered “very weak”, 0.10 < r ≤ 0.33 was considered “weak”, 0.33 < r ≤ 0.66 was considered “moderate” and r > 0.66 was considered “strong”.

The receiver operating characteristic (ROC) curve was used to assess the diagnostic value of any of the studied values in predicting admission to the ICU and risk of death due to COVID-19. The cutoff point was determined using the Youden index. A comparison of independent ROC curves was also done. For all tests, *p* < 0.05 was considered statistically significant.

## 3. Results

A total of 7789 patients were tested during the study period. Of these, 1783 were children, with only 381 found to be positive for COVID-19 upon admission to the ER. A total of 4625 were adults, with only 614 found to be positive for COVID-19 upon admission to ER. There were 1381 elders in total, and only 243 were positive for COVID-19 upon admission to ER.

### 3.1. Children

The total number of studied pediatric patients was 381. Among these patients, all (100%) presented data for CRP, 134 (35.17%) for LDH, 115 (30.18%) for CK, 159 (41.73%) for 25-OH vitamin D, 97 (25.46%) for ferritin and 276 (72.44%) were tested for HDL cholesterol.

Demographic data is presented in Table 1. The sex distribution was similar (M/F = 55.64%/44.36%). Most children presented mild symptomatology (*n* = 203, 53.28%). There were 14 (3.67%) cases with ICU admission, of which 7 (50%) were severe and 2 critical. There were three (0.79%) deceased patients. There were only two critical cases, however, neither succumbed. Median and IQR values for the quantifiable data of the pediatric patients can be observed in Table 2.

Regarding sex grouping, no differences were observed at the Kruskal–Wallis test, with the exception of ferritin, which is physiologically different between the two sexes. At the Chi^2^ test, no associations could be observed with regard to severity, ICU admission or death. All the data can be observed in Table 3.

With regard to severity, differences were observed at the Kruskal–Wallis test, with the exception of age and CK levels. At the Chi^2^ test, associations could be observed with regard to severity and both ICU admission or death. All the data can be observed in Table 4.

A further link between data was assessed using Spearman’s rank correlation test. A total of 15 (3.94%) out of 381 patients had available data on all the studied variables, as seen in Table 5. Considering that CK and age did not show differences at the Kruskal–Wallis test, they were omitted from this analysis.

The following correlations were observed: a moderate inverse relationship between CRP and HDL, a strong direct relationship between CRP and ICU admission, a moderate inverse relationship between ferritin and HDL, a strong direct relationship between ferritin and ICU admission, a moderate direct relationship between ferritin and deceased, a strong inverse relationship between HDL cholesterol and ICU admission and a moderate direct relationship between ICU admission and deceased.

### 3.2. Adults

The total number of studied adult patients was 614. Among these patients, 562 (91.53%) presented data for CRP, 536 (87.3%) for LDH, 426 (69.38%) for CK, 466 (75.9%) for 25-OH vitamin D, 389 (63.36%) for ferritin and 575 (93.65%) were tested for HDL cholesterol.

Demographic data is presented in Table 6. The sex distribution was similar (M/F = 52.12%/47.88%). Most patients presented mild symptomatology (*n* = 215, 35.02%). There were 81 (13.19%) cases with ICU admission, of which 33 (40.74%) were severe and 30 (37.04%) were critical. There were 53 (8.63%) deceased patients.

Median and IQR values for the quantifiable data of the pediatric patients can be observed in Table 7.

Regarding sex grouping, no differences were observed in the Kruskal–Wallis test, with the exception CK, which is physiologically different between the two sexes and 25-OH vitamin D. Males tended to have lower 25-OH vitamin D levels (median = 24.28, *p* = 0.0028). No associations could be observed from the Chi^2^ test with regard to severity, ICU admission or death. All the data can be observed in Table 8.

With regard to severity, differences were observed in the Kruskal–Wallis test, with the exception of age and CK levels. Associations could be observed from the Chi^2^ test with regard to severity and both ICU admission or death. All the data can be observed in Table 9, with Dunn’s post-hoc test results in Table 10.

A further link between data was assessed using Spearman’s rank correlation test. A total of 220 (35.83%) out of 614 patients had available data on all the studied variables, as seen in Table 11. Considering that CK and age did not show differences at the Kruskal–Wallis test, they were omitted from this analysis.

The following correlations were observed: a moderate direct relationship between CRP and LDH, a moderate inverse relationship between CRP and 25-OH vitamin D, a moderate direct relationship between CRP and ferritin, a weak inverse relationship between CRP and HDL cholesterol and a moderate direct relationship with both ICU admission and deceased, a moderate inverse relationship between LDH and 25-OH vitamin D, a weak direct relationship between LDH and ferritin and with both ICU admission and deceased, a weak inverse relationship between 25-OH vitamin D and ferritin, a weak direct relationship between 25-OH vitamin D and HDL cholesterol, a moderate direct relationship with both ICU admission and deceased, a weak inverse relationship between ferritin and HDL cholesterol, a weak direct relationship between ferritin and ICU admission, a moderate relationship with deceased, weak inverse relationships between HDL cholesterol and both ICU admission and deceased and a strong direct relationship between ICU admission and deceased.

### 3.3. Elders

The total number of studied elder patients was 243. Among these patients, 229 (94.24%) presented data for CRP, 205 (84.36%) for LDH, 193 (79.42%) for CK, 191 (78.6%) for 25-OH vitamin D, 195 (80.25%) for ferritin and 195 (80.25%) were tested for HDL cholesterol.

Demographic data is presented in Table 12. The sex distribution was similar (M/F = 42.39%/57.61%). Most patients presented severe symptomatology (*n* = 73, 30.04%). There were 112 (46.09%) cases with ICU admission, of which 43 (38.39%) were severe and 31 (33.04%) were critical. There were 61 (25.1%) deceased patients.

Median and IQR values for the quantifiable data of the pediatric patients can be observed in Table 13.

Regarding sex grouping, alongside CK and ferritin, which are physiologically different between the two sexes, a noticeable difference was observed for LDH, which was higher in males (median = 223, *p* = 0.0462). No associations could be observed from the Chi^2^ test with regard to severity, ICU admission or death. All the data can be observed in Table 14.

With regard to severity, differences were observed in the Kruskal–Wallis test, with the exception of age and 25-OH vitamin D levels. Associations could be observed from the Chi^2^ test with regard to severity and both ICU admission or death. All the data can be observed in Table 15, with Dunn’s post-hoc test results in Table 16.

A further link between data was assessed using Spearman’s rank correlation test. A total of 77 (31.69%) out of 243 patients had available data on all the studied variables, as seen in Table 17. Considering that age did not show a difference in the Kruskal–Wallis test, it was omitted from this analysis. In this group, CK differences were observed in the previous test and were included, however, no correlation was statistically significant. Regarding 25-OH vitamin D, no differences were observed in the previous test in elders, yet there were significant correlations with CRP (weak and inverse) and ICU admission (moderate and inverse).

Other important correlations that were observed: a moderate direct link between CRP and LDH; a moderate direct link between CRP and ferritin; a moderate inverse relationship between CRP and HDL cholesterol; a moderate direct relationship between CRP and ICU admission; a weak direct relationship between CRP and deceased; a moderate direct link between LDH and ferritin, ICU admission and deceased; a weak inverse relationship between ferritin and HDL cholesterol; a moderate direct relationship with ICU admission and a weak direct link with deceased. HDL cholesterol also correlated with moderately and inversely with ICU admission; ICU admission was linked moderately and directly with deceased.

### 3.4. Risk for ICU Admission

Details from the testing using the ROC curve analysis with regard to ICU admission are shown in Table 18. CK was excluded from this analysis, as, across all testing, its AUC (area under curve) was not statically significant. Graphically, this data can be seen in Appendix A. Observed differences are between the 25-OH vitamin D AUC of elders and both adults and children, being significantly smaller (0.631 vs. 0.838, 0.778, respectively) and ferritin where the AUC of children was significantly higher than both adults and elders (0.949 vs. 0.807, 0.735, respectively).

Regarding 25-OH vitamin D, this might be due to lower levels in elders overall, considering that their median was also lower than that of both adults and children (19.21 vs. 25.96, 23.44, respectively). Ferritin might be a better marker in children than in adults or elders, due to its high AUC.

### 3.5. Risk for Death

Details from the testing using the ROC curve analysis with regard to death are shown in Table 19. CK was excluded from this analysis, as, across all testing, its AUC was not statically significant. Graphically, this data can be seen in Appendix A. Observed differences are between the CRP AUC of adults and elders (0.893 vs. 0.669); the LDH AUC of children with both adults and elders (0.936 vs. 0.729, 0.740, respectively); the 25-OH vitamin D AUC of elders with both adults and children (0.613 vs. 0.883, 0.803, respectively); the HDL AUC of children and elders (0.858 vs. 0.623); and all ferritin AUCs (children > adults > elders: 0.995 > 0.931 > 0.665).

## 4. Discussion

The emergence of the global crisis due to the COVID-19 pandemic has led to measures such as social distancing, self-isolation, quarantine. Children and the elderly are important, as they present somewhat polarizing sides with regard to risk and severity. While children have been less studied, especially in the early stages, elders were observed to have a higher risk of severe illness and mortality. This effect has led to social disfunction in both cases [24,25], and has also affected the psychosocial institutions to which some of these patients cater to [26]

In order to manage COVID-19 patients and avoid complications, quick identification and the supply of the right treatments are essential. On the other hand, in its inception, the disease affected predominantly elders and patients with comorbidities [27]

Regarding clinical presentation, the proportion of asymptomatic children fell within the range previously reported by studies such as Ylimaz et al. [9], Say et al. [28] or Jat et al. [29], which was approximately one-third of total pediatric patients (36.75%). Contrarily, as age progressed, so did the severity of the cases, with only 16.12% being asymptomatic and even fewer (9.47%) patients older than 65.

It was documented that CRP values have a predictive role with regard to severity in adults, due to the immune system responding to the SARS-CoV-2 infection [30,31]. This was also observed in our study, as all three types of analysis showed a link between CRP levels and severity, ICU admission or death.

CRP is known to trigger a cytokine storm, which results in a massive release on pro-inflammatory cytokines, which can cause acute lung injury, resulting in an unfavorable prognosis, especially in elders [32]. In our studied elders, those with higher CRP were in the severe or critical categories. Generally, CRP was associated with ICU admission across all age groups; however, it only correlated with mortality for adults and elders.

In a previous study, it was established that CRP was higher in older children and adolescents, whose immune system is more developed, and, hence, more akin to that of adults’ and elders’ [33]. Following a retrospective analysis of a group of 3424 children in Indonesia, it was concluded that CRP levels may be a good indicator for the treatment and early identification of the disease in children [34].

The levels of CRP in relation to ICU admission presented a good AUC for all three age groups, with no significant differences. The cutoff values were: 22.51 for children, who exhibited sensitivity 85.71% and specificity 84.74%; 18.52 for adults, who exhibited sensitivity 79.1% and specificity 75.35%; and 26.57 for elders, who exhibited sensitivity 94.17% and specificity 61.11%.

For mortality, the results showed that the AUC of elders (0.669) was different that the adults’ (0.893), with children being in the middle (0.846), but not statistically different. Cutoff values were: 13.42 for children, who exhibited sensitivity 100% and specificity 62.96%; 54.93 for adults, who exhibited sensitivity 76.74% and specificity 94.99%; and 25.91 for elders, who exhibited sensitivity 94.44% and specificity 44.57%. A study focusing on adults found similar values in that age group [35].

Information on LDH implication was contradictory [34,36]. In our study, its levels correlated well with severity, ICU admission and death. For ICU admission, AUCs were similar, with the predictive values being: 288 for children, who exhibited sensitivity 80% and specificity 67.74%; 302 for adults, who exhibited sensitivity 78.69% and specificity 74.53%; and 184 for elders, who exhibited sensitivity 78.35% and specificity 56.48%.

In the case of mortality, where the AUC of children was higher than of both adults’ and elders’ (0.936 vs. 0.729, 0.740, respectively). The cutoff values were: 354 for children, who exhibited sensitivity 100% and specificity 87.12%; 294 for adults, who exhibited sensitivity 74.47% and specificity 68.1%; and 195 for elders, who exhibited sensitivity 85.45% and specificity 55.33%. LDH levels have been associated with clinical severity since the beginning of the pandemic, and many authors agree on and encourage its use as a clinical biomarker [37,38].

Although CK levels have been associated with COVID-19 in adults, its use as a potential biomarker remains controversial [39,40,41]. In our study, no association could be observed between this biomarker and the severity of cases. As depicted in the Appendix A; its AUC value was not statistically significant.

The modulation of immunological response is significantly influenced by 25-OH vitamin D [15]. Children with inadequate 25-OH vitamin D may experience worse clinical outcomes in Omicron subvariant BA 2 infections, according to research by Peng et al. [42]. In a study by Michael et al., supplementation of 25-OH vitamin D in adults and elders proved to decrease the risk of mechanical ventilation and clinical severity in COVID-19 [43]. The usefulness of 25-OH vitamin D supplementation was also observed in a randomized controlled trial done by Zurita-Cruz et al. [44]. Studies by di Filippo et al. showed a strict association of VD levels with blood GLU and BMI in COVID-19 patients on one hand, and that 25-OHD levels at admission strongly predicted the occurrence of worsening outcomes in COVID-19 independently of the disease severity at presentation. [45,46].

Our research demonstrated that patients’ levels of 25-OH vitamin D were inversely related to the presence and severity of clinical symptoms, as well as ICU admission and death. A difference in AUC’s was observed when analyzing ICU admission, with both adults’ and children’ being greater than the elders’ (0.631 vs. 0.838, 0.778, respectively). Cutoff values were: 17.07 for children, who exhibited sensitivity 81.82% and specificity 76.35%; 20.05 for adults, who exhibited sensitivity 83.05% and specificity 86.81%; and 22.44 for elders, who exhibited sensitivity 86.81% and specificity 35%.

25-OH vitamin D levels were observed to be similar across all elder severity groups, while in the adult and child groups there were differences according to severity. As such, for the risk of death, elder AUC was lower than that of children’s and adults’ (0.613 vs. 0.803, 0.883, respectively). Cutoff values were: 14.11 for children, who exhibited sensitivity 100% and specificity 78.98%; 17.34 for adults, who exhibited sensitivity 85% and specificity 84.74%; and 17.88 for elders, who exhibited sensitivity 54% and specificity 67.38%.

Ferritin is another biomarker that has been associated with severity in adults and is able to predict poor outcomes and prognosis, although its use is still somewhat contested [47,48,49]. In our patients, ferritin proved to be a good predictor of severity, ICU admission and death. Risk for ICU admission by studying ROC curves showed a higher AUC for children than for adults and elders (0.949 vs. 0.807, 0.735, respectively), with the cutoff values being: 211 for children, who exhibited sensitivity 100% and specificity 86.02%; 529 for adults, who exhibited sensitivity 60.78% and specificity 89.35%; and 133 for elders, who exhibited sensitivity 80.85% and specificity 60.4%.

For the study or mortality, all AUCs were considered different, as follows: children (0.995), adults (0.931) and elders (0.665). The cutoff values were: 415 for children, who exhibited sensitivity 100% and specificity 98.95%; 529 for adults, who exhibited sensitivity 88.57% and specificity 89.83%; and 155 for elders, who exhibited sensitivity 80% and specificity 52.86%.

The last biomarker that was studied was HDL cholesterol. A modified metabolic aspect has been observed in many COVID-19 patients, and this includes blood lipid levels. HDL cholesterol has been associated with poorer outcomes and more severe cases [21,50]. It has been suggested that it is involved in the regulation of innate immune response through interaction with ABCA1 or ABCG1, which negatively regulates T-cell activation and the expression of inflammatory mediators [49]. In our study, HDL correlated inversely with severity, ICU admission and death. It proved a good predictor for ICU admission, with all AUCs being similar. The following cutoff points were established: 47.75 for children, who exhibited sensitivity 85.71% and specificity 56.11%; 31.81 for adults, who exhibited sensitivity 71.6% and specificity 59.31%; and 33.17 for elders, who exhibited sensitivity 48.91% and specificity 80.68%.

With regard to its role as a mortality predictor, a difference was observed between the AUC of children and that of elders’ (0.858, 0.623), with adults’ in-between, without significant differences. Cutoff points were: 41.35 for children, who exhibited sensitivity 100% and specificity 67.4%; 32.03 for adults, who exhibited sensitivity 69.23% and specificity 60.04%; and 30.99 for elders, who exhibited sensitivity 39.13% and specificity 81.21%.

No differences due to sex, except physiological, were observed across all studied groups. The sex distribution was similar, with a slight increase in males for children and adults (55.64%, 52.12%, respectively) and a slight increase in females in the elder groups (57.61%).

### Limitations

As in all retrospective studies, this article presents several limitations. First and foremost: data distribution was heterogeneous, as not all patients had available data for all studied biomarkers. For example, only a small number of pediatric patients (3.94%) presented data on all studied biomarkers. Data was also missing with regard to multi system inflammatory syndrome, which is a serious condition sometimes linked to COVID-19, or comorbidities, which are usually more common among older patients.

Another possible limitation is the small sample of critical and deceased pediatric patients, which might skew the results. In some cases, severity was assessed after admission and this time difference may affect interpretation of results. Lastly, data on children was collated, which might not account for differences within the children group (i.e., infants vs. children vs. teenagers).

## 5. Conclusions

It the present study, several biomarkers proved useful for the analysis of severity, ICU admission and mortality. CRP is one of the most important biomarkers and correlated well with outcomes across all ages, with the calculated cutoff values as a mortality predictor higher for adults than elders. LDH correlated well with outcomes across all ages, especially in adults and elders. CK values were mostly within normal ranges, and cannot be considered as a biomarker in the assessment of the health of pediatric patients with COVID-19 and did not offer any correlation results. 25-OH vitamin D is also an important marker of severity, ICU admission and death, especially in adults, but less so in elders. Ferritin correlated the best among all studied markers with outcomes across all ages; however, it would seem more useful for pediatric patients and somewhat less useful for elders. HDL cholesterol also inversely correlated well across all studied groups with regard to severity, ICU admission and death, being similarly useful as a predictor.

## Figures and Tables

**Table 1 medicina-59-00877-t001:** Demographic data of pediatric patients.

Total = 381		*n*	%
Sex	M	212	55.64%
	F	169	44.36%
Symptomatology	Asymptomatic	140	36.75%
	Mild	203	53.28%
	Moderate	27	7.09%
	Severe	9	2.36%
	Critical	2	0.52%
ICU admission		14	3.67%
Deceased		3	0.79%

**Table 2 medicina-59-00877-t002:** Studied numerical characteristics of pediatric patients.

	Age	CRP	LDH	CK	25-OHD	FER	HDL
Median	7.97	8.10	268.50	107.00	23.44	98.48	49.75
IQR	12.32 (1.17–13.95)	15.22 (3.27–18.49)	91.5 (220.25–311.75)	56.5 (77–133.5)	15.69 (16.71–32.4)	117 (62–179)	22.01 (38.62–60.63)

CRP: C-reactive protein, LDH: lactate dehydrogenase, CK: creatinekinase, 25-OHD: 25-OH vitamin D, FER: ferritin, HDL: high-density lipoprotein cholesterol, IQR: interquartile range

**Table 3 medicina-59-00877-t003:** The results of Kruskal–Wallis and Chi^2^ testing with regard to sex in pediatric patients.

Sex		M	F	*p*
Age	Median	6.21	8.37	0.1453
	IQR	12.04 (0.97–13.01)	11.58 (2.48–14.06)	
CRP	Median	8.10	8.29	0.7403
	IQR	15.9 (2.87–18.76)	14.22 (3.71–17.93)	
LDH	Median	271.5	262.5	0.5491
	IQR	86 (227–313)	94.5 (216.5–311)	
CK	Median	99	109	0.9679
	IQR	59.25 (76.25–135.5)	35 (87.5–122.5)	
25-OHD	Median	22.76	24.52	0.8828
	IQR	17.12 (16.51–33.64)	14.55 (16.78–31.33)	
FER	Median	122	76	0.0047
	IQR	116 (72.5–188.5)	52.5 (47.5–100)	
HDL	Median	48.47	50.5	0.6192
	IQR	24.72 (36.33–61.04)	19.74 (40.79–60.53)	
Severity *	Asymptomatic	76 (19.95%)	64 (16.80%)	0.9605
	Mild	114 (29.92%)	90 (23.62%)	
	Moderate	15 (3.94%)	11 (2.89%)	
	Severe	6 (1.57%)	3 (0.79%)	
	Critical	1 (0.26%)	1 (0.26%)	
ICU admission *		11 (2.89%)	7 (1.84%)	0.6328
Deceased *		1 (0.26%)	2 (0.52%)	0.4355

*: Chi^2^ test.

**Table 4 medicina-59-00877-t004:** The results of Kruskal–Wallis and Chi^2^ testing with regard to severity in pediatric patients.

Severity		Asymptomatic (1)	Mild (2)	Moderate (3)	Severe (4)	*p*	Dunn’s
Age	Median	6.3	7.94	7.11	9.08	0.1761	
	IQR	11.64 (0.91–12.55)	12.19 (1.75–13.94)	13.74 (0.72–14.46)	3.97 (8.18–12.15)		
CRP	Median	2.9	12.23	39.11	66.93	<0.0001	1 vs. 2, 3, 4; 2 vs. 3, 4
	IQR	3.84 (1.13–4.97)	13.47 (6.48–19.95)	44.63 (14.56–59.19)	33.85 (52.03–85.88)		
LDH	Median	218	273	341	369	<0.0001	1 vs. 2, 3, 4; 2 vs. 3, 4
	IQR	43 (207–250)	70 (240–310)	67.5 (299–366.5)	100 (295.5–395.5)		
CK	Median	87	112	112	125.5	0.0625	
	IQR	61.25 (64.75–126)	62 (79–141)	33 (92–125)	19 (115–134)		
25-OHD	Median	30.56	21.8	17.67	13.47	0.0023	1 vs. 2, 4
	IQR	22.89 (18–40.89)	11.78 (17.45–29.23)	26.95 (8.72–35.67)	6.85 (11.79–18.64)		
FER	Median	99	81.5	124	279	0.0086	1 vs. 4; 2 vs. 4
	IQR	71 (68–139)	129 (53–182)	106 (53.5–159.5)	453.75 (248.75–702.5)		
HDL	Median	50.07	49.47	53.29	31.11	0.0197	4 vs. 1, 2, 3
	IQR	20.75 (40.07–60.82)	22.56 (38.44–61)	25.02 (39.75–64.77)	18.16 (23.29–41.45)		
ICU admission *		0 (0.00%)	8 (2.10%)	6 (1.57%)	4 (1.05%)	<0.0001	
Deceased *		0 (0.00%)	1 (0.26%)	0 (0.00%)	2 (0.52%)	<0.0001	

*: Chi^2^ test.

**Table 5 medicina-59-00877-t005:** The results of the Spearman’s rank correlation test in pediatric patients.

	CRP	LDH	25-OHD	FER	HDL	ICU	Deceased
CRP	1	0.4576 *p* = 0.0864	−0.4821 *p* = 0.0739	0.345 *p* = 0.2080	−0.6 *p* = 0.0214 *	0.663 *p* = 0.0071 *	0.4539 *p* = 0.0892
LDH	0.4576 *p* = 0.0864	1	−0.3414 *p* = 0.213	0.4401 *p* = 0.1007	−0.3271 *p* = 0.2341	0.3492 *p* = 0.202	0.4543 *p* = 0.0889
25-OHD	−0.4821 *p* = 0.0739	−0.3414 *p* = 0.213	1	−0.2663 *p* = 0.3374	0.4679 *p* = 0.0832	−0.4885 *p* = 0.0647	−0.3631 *p* = 0.1834
FER	0.345 *p* = 0.2080	0.4401 *p* = 0.1007	−0.2663 *p* = 0.3374	1	−0.6434 *p* = 0.0097 *	0.6985 *p* = 0.0038 *	0.5906 *p* = 0.0204 *
HDL	−0.6 *p* = 0.0214 *	−0.3271 *p* = 0.2341	0.4679 *p* = 0.0832	−0.6434 *p* = 0.0097 *	1	−0.7676 *p* = 0.0008 *	−0.4993 *p* = 0.0581
ICU	0.663 *p* = 0.0071 *	0.3492 *p* = 0.202	−0.4885 *p* = 0.0647	0.6985 *p* = 0.0038 *	−0.7676 *p* = 0.0008 *	1	0.6504 *p* = 0.0087 *
Deceased	0.4539 *p* = 0.0892	0.4543 *p* = 0.0889	−0.3631 *p* = 0.1834	0.5906 *p* = 0.0204 *	−0.4993 *p* = 0.0581	0.6504 *p* = 0.0087 *	1

*: *p* < 0.05, statistically significant.

**Table 6 medicina-59-00877-t006:** Demographic data of adult patients.

Total = 614		*n*	%
Sex	M	320	52.12%
	F	294	47.88%
Symptomatology	Asymptomatic	99	16.12%
	Mild	215	35.02%
	Moderate	152	24.76%
	Severe	60	9.77%
	Critical	88	14.33%
ICU admission		81	13.19%
Deceased		53	8.63%

**Table 7 medicina-59-00877-t007:** Studied numerical characteristics of adult patients.

	Age	CRP	LDH	CK	25-OHD	FER	HDL
Median	42.63	9.95	258.00	264.50	25.96	209.00	34.56
IQR	21.42 (31.78–53.2)	20.49 (5.05–25.54)	108 (220–328)	217 (162.75–379.75)	14.94 (19.22–34.15)	365 (124–489)	18.58 (25.72–44.3)

**Table 8 medicina-59-00877-t008:** The results of Kruskal–Wallis and Chi^2^ testing with regard to sex in Adults patients.

Sex		M	F	*p*
Age	Median	44.77	39.79	0.0526
	IQR	16.79 (35.27–52.06)	23.08 (30.3–53.38)	
CRP	Median	11.43	9.59	0.2423
	IQR	26.17 (5.04–31.21)	18.83 (5.22–24.05)	
LDH	Median	272	261	0.2519
	IQR	105.25 (227–332.25)	123.75 (215.5–339.25)	
CK	Median	324	215	<0.0001
	IQR	213.5 (196.5–410)	189 (139–328)	
25-OHD	Median	24.28	26.85	0.0028
	IQR	13.95 (17.52–31.47)	16.07 (19.45–35.52)	
FER	Median	213	185	0.0987
	IQR	273.75 (133–406.75)	203 (121–324)	
HDL	Median	33.85	35.64	0.1629
	IQR	17.43 (25.37–42.8)	16.86 (27.6–44.47)	
Severity *	Asymptomatic	40 (6.51%)	59 (9.61%)	0.1956
	Mild	49 (7.98%)	39 (6.35%)	
	Moderate	97 (15.80%)	118 (19.22%)	
	Severe	79 (12.87%)	73 (11.89%)	
	Critical	29 (4.72%)	31 (5.05%)	
ICU admission *		39 (6.35%)	42 (6.84%)	0.4432
Deceased *		26 (4.23%)	27 (4.40%)	0.6410

*: Chi^2^ test.

**Table 9 medicina-59-00877-t009:** The results of Kruskal–Wallis and Chi^2^ testing with regard to severity in Adults patients.

Severity		Asymptomatic	Mild	Moderate	Severe	Critical	*p*
Age	Median	41.91	42.52	42.28	43.89	43.46	0.1761
	IQR	22.11 (31.53–53.64)	20.13 (33.23–53.36)	18.44 (33.85–52.29)	22.14 (33.22–55.35)	19.94 (31.46–51.4)	
CRP	Median	5.74	6.18	18.61	58.65	33.31	<0.0001
	IQR	5.23 (2.8–8.03)	8.53 (2.62–11.15)	25.94 (8.69–34.63)	35.74 (34.51–70.25)	42.01 (17.15–59.16)	
LDH	Median	204	245.5	287	324	390.5	<0.0001
	IQR	70 (168–238)	66.5 (214.5–281)	96 (239.75–335.75)	154.75 (278.75–433.5)	83.5 (359–442.5)	
CK	Median	327	269	277	227	275	0.0625
	IQR	177 (197–374)	191 (161–352)	244 (159.75–403.75)	272.25 (108.75–381)	244 (191–435)	
25-OHD	Median	37.6	28.87	22.96	12.32	11.17	0.0023
	IQR	19.51 (29.88–49.39)	12.59 (22.65–35.23)	10.67 (17.82–28.49)	15.19 (8.32–23.51)	11.6 (7.73–19.33)	
FER	Median	135	146.5	319	557.5	381	0.0086
	IQR	143.75 (57.75–201.5)	232 (85–317)	266.5 (227–493.5)	1180 (239.5–1419.5)	913.75 (234.25–1148)	
HDL	Median	35.74	36.59	34.15	30.9	29.36	0.0197
	IQR	18.71 (27.6–46.31)	15.6 (28.84–44.44)	19.91 (25.7–45.61)	17.57 (22.27–39.84)	13.51 (22.43–35.94)	
ICU admission *		3 (0.49%)	5 (0.81%)	10 (1.63%)	33 (5.37%)	30 (4.89%)	<0.0001
Deceased *		1 (0.16%)	2 (0.33%)	3 (0.49%)	29 (4.72%)	18 (2.93%)	<0.0001

*: Chi^2^ test.

**Table 10 medicina-59-00877-t010:** The results of Dunn’s post-hoc testing with regard to severity in Adults patients.

Dunn’s Post-hoc Test	Asymptomatic (1)	Mild (2)	Moderate (3)	Severe (4)	Critical (5)	*p*
Age	-					0.9516
CRP	1 vs. 3, 4, 5; 2 vs. 3, 4, 5; 3 vs. 4, 5					<0.0001
LDH	1 vs. 2, 3, 4, 5; 2 vs. 3, 4, 5; 3 vs. 5					<0.0001
CK	-					0.0767
25-OHD	1 vs. 2, 3, 4, 5; 2 vs. 3, 4, 5; 3 vs. 4, 5					<0.0001
FER	1 vs. 3, 4, 5; 2 vs. 3, 4, 5					<0.0001
HDL	1 vs. 5; 2 vs. 4, 5; 3 vs. 5					<0.0001

**Table 11 medicina-59-00877-t011:** The results of the Spearman’s rank correlation test in Adults patients.

	CRP	LDH	25-OHD	FER	HDL	ICU	Deceased
CRP	1	0.3464 *p* < 0.0001 *	−0.481 *p* < 0.0001 *	0.3801 *p* < 0.0001 *	−0.1474 *p* = 0.0288 *	0.3632 *p* < 0.0001 *	0.3702 *p* < 0.0001 *
LDH	0.3464 *p* < 0.0001 *	1	−0.3585 *p* < 0.0001 *	0.2773 *p* < 0.0001 *	−0.1242 *p* = 0.0659	0.3011 *p* < 0.0001 *	0.2589 *p* = 0.0001 *
25-OHD	−0.481 *p* < 0.0001 *	−0.3585 *p* < 0.0001 *	1	−0.293 *p* < 0.0001 *	0.1497 *p* = 0.0264 *	−0.4051 *p* < 0.0001 *	−0.4027 *p* < 0.0001 *
FER	0.3801 *p* ≤ 0.0001 *	0.2773 *p* ≤ 0.0001 *	−0.293 *p* ≤ 0.0001 *	1	−0.2143 *p* = 0.0014 *	0.3107 *p* ≤ 0.0001 *	0.3891 *p* ≤ 0.0001 *
HDL	−0.1474 *p* = 0.0288 *	−0.1242 *p* = 0.0659	0.1497 *p* = 0.0264 *	−0.2143 *p* = 0.0014 *	1	−0.2975 *p* < 0.0001 *	−0.3221 *p* < 0.0001 *
ICU	0.3632 *p* < 0.0001 *	0.3011 *p* < 0.0001 *	−0.4051 *p* < 0.0001 *	0.3107 *p* < 0.0001 *	−0.2975 *p* < 0.0001 *	1	0.8363 *p* < 0.0001 *
Deceased	0.3702 *p* < 0.0001 *	0.2589 *p* = 0.0001 *	−0.4027 *p* < 0.0001 *	0.3891 *p* < 0.0001 *	−0.3221 *p* < 0.0001 *	0.8363 *p* < 0.0001 *	1

*: *p* < 0.05, statistically significant.

**Table 12 medicina-59-00877-t012:** Demographic data of elder patients.

Total = 243		*n*	%
Sex	M	103	42.39%
	F	140	57.61%
Symptomatology	Asymptomatic	23	9.47%
	Mild	45	18.52%
	Moderate	55	22.63%
	Severe	73	30.04%
	Critical	47	19.34%
ICU admission		112	46.09%
Deceased		61	25.10%

**Table 13 medicina-59-00877-t013:** Studied numerical characteristics of elder patients.

	Age	CRP	LDH	CK	25-OHD	FER	HDL
Median	72.65	8.10	268.50	107.00	23.44	98.48	49.75
IQR	11.49 (69.18–80.68)	50.43 (13.8–64.22)	185 (140–325)	116 (80–196)	7.54 (14.86–22.39)	382 (68–450)	13.41 (31.23–44.63)

**Table 14 medicina-59-00877-t014:** The results of Kruskal–Wallis and Chi^2^ testing with regard to sex in elder patients.

Sex		M	F	*p*
Age	Median	76.73	72.25	0.0928
	IQR	10.11 (70.28–80.39)	13.71 (68.39–82.1)	
CRP	Median	40.39	37.22	0.8685
	IQR	46.81 (14.88–61.69)	54.88 (13.63–68.51)	
LDH	Median	223	201.5	0.0462
	IQR	197.75 (152.5–350.25)	173.5 (119–292.5)	
CK	Median	158	131.5	0.0306
	IQR	153.75 (86–239.75)	112 (74–186)	
25-OHD	Median	19.37	18.91	0.1983
	IQR	6.21 (17.08–23.29)	9.72 (12.51–22.23)	
FER	Median	282	149.5	0.0024
	IQR	390 (103.75–493.75)	310.5 (24–334.5)	
HDL	Median	36.85	39.02	0.2193
	IQR	13.23 (30.96–44.19)	13.73 (31.29–45.02)	
Severity *	Asymptomatic	14 (5.76%)	9 (3.70%)	0.6813
	Mild	30 (12.35%)	17 (7.00%)	
	Moderate	28 (11.52%)	17 (7.00%)	
	Severe	29 (11.93%)	26 (10.70%)	
	Critical	39 (16.05%)	34 (13.99%)	
ICU admission *		44 (18.11%)	68 (27.98%)	0.0580
Deceased *		26 (10.70%)	35 (14.40%)	0.9657

*: Chi^2^ test.

**Table 15 medicina-59-00877-t015:** The results of Kruskal–Wallis and Chi^2^ testing with regard to severity in elder patients.

Severity		Asymptomatic	Mild	Moderate	Severe	Critical	*p*
Age	Median	72.65	70.68	75.18	72.27	76.93	0.0771
	IQR	11.76 (68.84–80.6)	6.57 (68.36–74.93)	11.04 (70.78–81.82)	12.94 (68.87–81.81)	11.21 (70.39–81.6)	
CRP	Median	4.57	12.38	27.59	61.95	57.5	<0.0001
	IQR	6.7 (2.46–9.16)	20.05 (6.19–26.24)	29.67 (16.61–46.28)	33.43 (40.51–73.93)	41.14 (36.01–77.15)	
LDH	Median	101	167.5	218	256	292	<0.0001
	IQR	110 (49–159)	112.5 (116–228.5)	217.75 (99–316.75)	178.5 (169.5–348)	303.25 (193.25–496.5)	
CK	Median	75	139	144	156.5	145	0.0196
	IQR	105.75 (37–142.75)	142 (88–230)	113.75 (80.5–194.25)	121 (91–212)	133 (80.75–213.75)	
25-OHD	Median	23.45	20.82	20.06	18.9	18.43	0.0802
	IQR	19.2 (16.1–35.3)	11.72 (14.39–26.11)	4.26 (17.34–21.61)	6.83 (14.91–21.74)	9.35 (12.53–21.87)	
FER	Median	13.5	16	163	453	403	<0.0001
	IQR	63.5 (7.5–71)	37.5 (11.25–48.75)	149.5 (83–232.5)	325.75 (263.5–589.25)	473 (258.5–731.5)	
HDL	Median	43.2	42.39	38.19	34.29	32.22	<0.0001
	IQR	15.59 (37.31–52.9)	13.62 (35.13–48.75)	11.36 (33.01–44.37)	13.54 (29.33–42.87)	13.02 (26.13–39.15)	
ICU admission *		0 (0.00%)	9 (1.47%)	23 (3.75%)	43 (7.00%)	37 (6.03%)	<0.0001
Deceased *		2 (0.33%)	5 (0.81%)	15 (2.44%)	22 (3.58%)	17 (2.77%)	0.0162

*: Chi^2^ test.

**Table 16 medicina-59-00877-t016:** The results of Dunn’s post-hoc testing with regard to severity in elder patients.

Dunn’s Post-Hoc Test	Asymptomatic (1)	Mild (2)	Moderate (3)	Severe (4)	Critical (5)	*p*
Age	-					0.0771
CRP	1 vs. 3,4,5; 2 vs. 4,5; 3 vs. 4,5					<0.0001
LDH	1 vs. 3,4,5; 2 vs. 4,5					<0.0001
CK	1 vs. 2,4					0.0196
25-OHD	-					0.0802
FER	1 vs. 3,4,5; 2 vs. 3,4,5; 3 vs. 4,5					<0.0001
HDL	1 vs. 4,5					<0.0001

**Table 17 medicina-59-00877-t017:** The results of the Spearman’s rank correlation test in elder patients.

	CRP	LDH	CK	25-OHD	FER	HDL	ICU	Deceased
CRP	1	0.4076 *p* = 0.0002 *	0.2104 *p* = 0.0662	−0.2577 *p* = 0.0236 *	0.5168 *p* < 0.0001 *	−0.4595 *p* < 0.0001 *	0.5738 *p* < 0.0001 *	0.2375 *p* = 0.0376 *
LDH	0.4076 *p* = 0.0002 *	1	0.2139 *p* = 0.0618	−0.2024 *p* = 0.0776	0.5117 *p* < 0.0001 *	−0.1662 *p* = 0.1486	0.4706 *p* < 0.0001 *	0.4218 *p* = 0.0001 *
CK	0.2104 *p* = 0.0662	0.2139 *p* = 0.0618	1	−0.0172 *p* = 0.8820	0.2095 *p* = 0.0675	0.1491 *p* = 0.1955	0.0957 *p* = 0.4080	0.2093 *p* = 0.0678
25-OHD	−0.2577 *p* = 0.0236 *	−0.2024 *p* = 0.0776	−0.0172 *p* = 0.8820	1	−0.0617 *p* = 0.5943	0.1727 *p* = 0.1331	−0.3697 *p* = 0.0009 *	−0.143 *p* = 0.2147
FER	0.5168 *p* < 0.0001 *	0.5117 *p* < 0.0001 *	0.2095 *p* = 0.0675	−0.0617 *p* = 0.5943	1	−0.2888 *p* = 0.0108 *	0.3873 *p* = 0.0005 *	0.2532 *p* = 0.0263 *
HDL	−0.4595 *p* < 0.0001 *	−0.1662 *p* = 0.1486	0.1491 *p* = 0.1955	0.1727 *p* = 0.1331	−0.2888 *p* = 0.0108 *	1	−0.4365 *p* < 0.0001 *	−0.021 *p* = 0.8562
ICU	0.5738 *p* < 0.0001 *	0.4706 *p* < 0.0001 *	0.0957 *p* = 0.4080	−0.3697 *p* = 0.0009 *	0.3873 *p* = 0.0005 *	−0.4365 *p* < 0.0001 *	1	0.4419 *p* < 0.0001 *
Deceased	0.2375 *p* = 0.0376 *	0.4218 *p* = 0.0001 *	0.2093 *p* = 0.0678	−0.143 *p* = 0.2147	0.2532 *p* = 0.0263 *	−0.021 *p* = 0.8562	0.4419 *p* < 0.0001 *	1

*: *p* < 0.05, statistically significant.

**Table 18 medicina-59-00877-t018:** Results of the ROC curve analysis and comparison with regard to ICU admission.

ICU		CRP	LDH	25-OHD	FER	HDL
Adults (1)	AUC (95% CI)	0.810 (0.776–0.842)	0.792 (0.755–0.825)	0.838 (0.801–0.870)	0.807 (0.764–0.845)	0.719 (0.680–0.755)
	Cutoff	>18.52	>302	≤20.5	>529	≤32.81
	Sensitivity; Specificity	Se: 79.1; Sp: 75.35	Se: 78.69; Sp: 74.53	Se: 83.05; Sp: 73.71	Se: 60.78; Sp: 89.35	Se: 71.6; Sp: 59.31
Children (2)	AUC (95% CI)	0.899 (0.864–0.927)	0.793 (0.715–0.858)	0.778 (0.705–0.840)	0.949 (0.884–0.983)	0.718 (0.661–0.770)
	Cutoff	>22.51	>288	≤17.07	>211	≤47.75
	Sensitivity; Specificity	Se: 85.71; Sp: 84.74	Se: 80; Sp: 67.74	Se: 81.82; Sp: 76.35	Se: 100; Sp: 86.02	Se: 85.71; Sp: 56.11
Elders (3)	AUC (95% CI)	0.791 (0.732–0.842)	0.727 (0.660–0.786)	0.631 (0.558–0.699)	0.735 (0.667–0.796)	0.672 (0.602–0.738)
	Cutoff	>26.57	>184	≤22.44	>133	≤33.17
	Sensitivity; Specificity	Se: 94.17; Sp: 61.11	Se: 78.35; Sp: 56.48	Se: 86.81; Sp: 35	Se: 80.85; Sp: 60.4	Se: 48.91; Sp: 80.58
Pairwise comparison ROC with regard to ICU admission (*p*)	1 vs. 2	0.1391	0.9853	0.2718	0.0046 *	0.9925
	1 vs. 3	0.6513	0.1870	<0.0001 *	0.1543	0.3380
	2 vs. 3	0.0680	0.3743	0.0175 *	<0.0001 *	0.5662

*: *p* < 0.05, statistically significant.

**Table 19 medicina-59-00877-t019:** Results of the ROC curve analysis and comparison with regard to death.

Death		CRP	LDH	25-OHD	FER	HDL
Adults (1)	AUC (95% CI)	0.893 (0.864–0.917)	0.729 (0.689–0.766)	0.883 (0.850–0.911)	0.931 (0.902–0.954)	0.692 (0.652–0.729)
	Cutoff	>54.93	>294	≤17.34	>529	≤32.03
	Sensitivity; Specificity	Se: 76.74; Sp: 94.99	Se: 74.47; Sp: 68.1	Se: 85; Sp: 84.74	Se: 88.57; Sp: 89.83	Se: 69.23; Sp: 60.04
Children (2)	AUC (95% CI)	0.846 (0.805–0.880)	0.936 (0.880–0.971)	0.803 (0.732–0.861)	0.995 (0.953–1.000)	0.858 (0.811–0.897)
	Cutoff	>13.42	>354	≤14.11	>415	≤41.35
	Sensitivity; Specificity	Se: 100; Sp: 62.96	Se: 100; Sp: 87.12	Se: 100; Sp: 78.98	Se: 100; Sp: 98.95	Se: 100; Sp: 67.4
Elders (3)	AUC (95% CI)	0.669 (0.604–0.729)	0.740 (0.674–0.798)	0.613 (0.540– 0.68)	0.665 (0.594–0.731)	0.623 (0.551–0.691)
	Cutoff	>25.91	>195	≤17.88	>155	≤30.99
	Sensitivity; Specificity	Se: 94.44; Sp: 44.57	Se: 85.45; Sp: 55.33	Se: 54; Sp: 67.38	Se: 80; Sp: 52.86	Se: 39.13; Sp: 81.21
Pairwise comparison ROC with regard to death (*p*)	1 vs. 2	0.6766	0.0085 *	0.0752	0.0256 *	0.1009
	1 vs. 3	<0.0001 *	0.8512	<0.0001 *	<0.0001 *	0.2464
	2 vs. 3	0.1230	0.0100 *	0.0009 *	<0.0001 *	0.0267 *

*: *p* < 0.05, statistically significant.

## Data Availability

Data available on request.

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
