# Peer review of "COVID-19 Biomarkers Comparison: Children, Adults and Elders"

_medicina, 2023, doi:10.3390/medicina59050877_

Round 1

Reviewer 1 Report

Dear Editor,

  thanks for the opportunity to revise this work entitled "COVID-19 biomarkers comparison: children, adults and elders" proposed by Dr. Capraru.   The work is very interesting. The authors aimed to evaluate the influence of prognostic biochemical markers in COVID-19 patients, and, so interestingly, analysing their different impacts on the basis of different populations studied: elders, adults and childrens.   The authors mainly confirmed the prognostic role of these markers also expanding their possible utility in clinical practice.    Nevertheless, i have several comments that may improve the manuscript quality.   Introduction:  i suggest to directly focus the paper attention on COVID-19 without comments or intro SARS-CoV infections; line 55, please delete can define the abbreviations in brackets after the extended name i suggest to use in the entire text the name "25(OH) vitamin D" and not only "vitamin D", that could be misunderstood line 73, i suggest to remove "few". A large amount of data were previously published regarding vitamin D and COVID-19. In particular, i suggest to comment in intro or discussion section a previous retrospective study observing strict relationships between 25(OH) vitamin D levels and proinflammatory markers (as well in your study, doi: 10.1210/clinem/dgab599.), and, since the data on vitamin D and COVID-19 patients are mainly retrospective with possible bias, please comment also novel prospective data confirming the negative influence of low vitamin D levels in these patients (DOI: 10.1007/s12020-023-03331-9).  line 82: please, change statistics with epidemiological data   Methods data collection, please report the greater/less-than signs in the references ranges.    Results line 153, please, change persons in patients. Please, perform a careful english revision.   291, median and not medina   thanks

.

Author Response

Answer to Reviewer 1

Thank you very much for reviewing our article and for all your recommendations, we took all of them into consideration, and now we hope that the quality and the clarity of the present paper was improved, making it a more valuable one.

In response, here are punctually our answers to your recommendations.

  1. Introduction: i suggest to directly focus the paper attention on COVID-19 without comments or intro SARS-CoV infections

Thank you. This was addressed.

  1. line 55, please delete can define the abbreviations in brackets after the extended name i suggest to use in the entire text the name "25(OH) vitamin D" and not only "vitamin D", that could be misunderstood

Thank you. This was addressed.

  1. line 73, i suggest to remove "few". A large amount of data were previously published regarding vitamin D and COVID-19. In particular, i suggest to comment in intro or discussion section a previous retrospective study observing strict relationships between 25(OH) vitamin D levels and proinflammatory markers (as well in your study, doi: 10.1210/clinem/dgab599.) and, since the data on vitamin D and COVID-19 patients are mainly retrospective with possible bias, please comment also novel prospective data confirming the negative influence of low vitamin D levels in these patients (DOI: 10.1007/s12020-023-03331-9).

The phrase has been edited as such: “Although there were few studies in the beginning, a link between symptomatology, severity, and outcome in COVID-19 in regards to 25-OH vitamin D has been observed”. The recommended citation has been added in the discussion part.

  1. line 82: please, change statistics with epidemiological data

Thank you. This was addressed.

  1. Methods data collection, please report the greater/less-than signs in the references ranges.

Thank you. This was addressed.

  1. Results line 153, please, change persons in patients. Please, perform a careful english revision.

Thank you. This was addressed.

  1. 291, median and not medina

Thank you. This was addressed.

Best regards,

Thank you for your comments and recommendations. They have been addressed in the attached word file. 

Reviewer 2 Report

Comments:

1. Could the authors please provide the ethical approval for their research study please?

2. Could the authors please justify the numbers of study participants i.e., how was their study powered?

As above.

Author Response

Answer to Reviewer 2

Thank you very much for reviewing our article and for all your recommendations, we took all of them into consideration, and now we hope that the quality and the clarity of the present paper was improved, making it a more valuable one.

In response, here are punctually our answers to your recommendations.

  1. Could the authors please provide the ethical approval for their research study please?

Thank you, the following number has been provided and was also written in the manuscript: 98/2022

  1. Could the authors please justify the numbers of study participants i.e., how was their study powered?

Thank you. The following has been added: “Sample size was calculated using the G*Power software (v 3.1.9.6), using an a priori test to calculate the minimum sample size for a medium effect size (d= 0.25), a power of 80% and an alpha of 0.05. The result indicated that a minimum of 270 patients were needed.”

Best regards,

Thank you for your comments and recommendations. They have been addressed in the attached word file. 

Reviewer 3 Report

Coronavirus usually cause a cold or mild respiratory symptoms while pathogenic zoonotic coronavirus such as SARS-CoV or MERS cause severe diseases. Authors need to clarify its introduction.
Have you undergone an ethics review? Although it uses past data, I think ethical review might be necessary.
The findings of blood test was upon admission while severity was diagnosed after admission. This time difference may affect interpretation of result. Authors need to discuss in this point.
Is it possible to collate blood test results in children from zero to 18 years?
What is severity/critical =1 in table 3?
The title of ROC curve should clearly indicate that the ICU is being evaluated.

Ldh, even though I see you the same. The specificity is completely different. Authors carefully discuss LOC curve not only with AUC but behaviour of Sensi/Speci

Author Response

Answer to Reviewer 3

Thank you very much for reviewing our article and for all your recommendations, we took all of them into consideration, and now we hope that the quality and the clarity of the present paper was improved, making it a more valuable one.

In response, here are punctually our answers to your recommendations.

  1. Coronavirus usually cause a cold or mild respiratory symptoms while pathogenic zoonotic coronavirus such as SARS-CoV or MERS cause severe diseases. Authors need to clarify its introduction.

Thank you, the mention of SARS and MERS has been removed, as it was requested by another reviewer.

  1. Have you undergone an ethics review? Although it uses past data, I think ethical review might be necessary.

Thank you, the following number has been provided and was also written in the manuscript: 98/2022

  1. The findings of blood test was upon admission while severity was diagnosed after admission. This time difference may affect interpretation of result. Authors need to discuss in this point.

Thank you this has been added in the limitations section: “In some cases, severity was assessed after admission and this time difference may affect interpretation of results.”

  1. Is it possible to collate blood test results in children from zero to 18 years?

Thank you for noticing this. This was done in order to make data processing easier. We actually did a study only on children, which observed a difference between children younger than 1 year and older than 1 year for CRP and LDH, but not the rest  (doi: 10.3390/life13010091). In order to be more transparent, this was added as a limitation: “Lastly, data on children was collated, which might not account for differences within the children group (i.e. infants vs children vs teenagers).”

  1. What is severity/critical =1 in table 3?

Thank you for noticing this mistake we overlook. That was data from a very early draft. It was corrected now.

  1. The title of ROC curve should clearly indicate that the ICU is being evaluated.

Thank you, this was addressed.

  1. Ldh, even though I see you the same. The specificity is completely different. Authors carefully discuss LOC curve not only with AUC but behaviour of Sensi/Speci

Thank you, this was addressed.

Best regards,

Iulia Cristina Bagiu

Thank you for your comments and recommendations. They have been addressed in the attached word file. 

Round 2

Reviewer 2 Report

The authors have satisfactorily responded to my comments.

As above.